# Preparation, Multispectroscopic Characterization, and Stability Analysis of *Monascus* Red Pigments—Whey Protein Isolate Complex

**DOI:** 10.3390/foods12091745

**Published:** 2023-04-23

**Authors:** Huafa Lai, Jiahao Wang, Shengjia Liao, Gang Liu, Liling Wang, Yi He, Chao Gao

**Affiliations:** 1National R&D Center for Se-Rich Agricultural Products Processing, Hubei Engineering Research Center for Deep Processing of Green Se-Rich Agricultural Products, School of Modern Industry for Selenium Science and Engineering, Wuhan Polytechnic University, Wuhan 430023, China; laihuafa123@126.com (H.L.); wangjiahao0307@163.com (J.W.); gaochao@whpu.edu.cn (C.G.); 2Key Laboratory for Deep Processing of Major Grain and Oil, Ministry of Education, Hubei Key Laboratory for Processing and Transformation of Agricultural Products, School of Food Science and Engineering, Wuhan Polytechnic University, Wuhan 430023, China; 15271375954@163.com (S.L.); liugang1982@whpu.edu.cn (G.L.); 3College of Food Science and Engineering, Tarim University, Alar 843300, China; 120060036@taru.edu.cn

**Keywords:** whey protein isolate, *Monascus* red pigments, combination constants, stability analysis

## Abstract

*Monascus* red pigments (MRPs) are mainly used as natural food colorants; however, their application is limited due to their poor stability. To expand their areas of application, we investigated the binding constants and capacity of MRPs to whey protein isolate (WPI) and whey protein hydrolysate (WPH) and calculated the surface hydrophobicities of WPI and WPH. MRPs were combined with WPI and WPH at a hydrolysis degree (DH) of 0.5% to form the complexes (DH = 0.0%) and (DH = 0.5%), respectively. Subsequently, the structural characteristics of complex (DH = 0.5%) and WPI were characterized and the color retention rates of both complexes and MRPs were investigated under different pretreatment conditions. The results showed that the maximum binding constant of WPI with MRPs was 0.670 ± 0.06 U^−1^ and the maximum binding capacity was 180 U/g. Furthermore, the thermal degradation of complex (DH = 0.0%), complex (DH = 0.5%), and MRPs in a water bath at 50–100 °C followed a first-order kinetic model. Thus, the interaction of WPI with MRPs could alter the protein conformation of WPI and effectively protect the stability of MRPs.

## 1. Introduction

*Monascin* (MS), identified from *Monascus*-fermented products, is a yellow lipid-soluble azaphilonoid pigment with promising biological activities [1]. *Monascus* pigments (MPs) have been used in the food industry for more than 2000 years and are known for their safety and bold coloring [2]. Therefore, MPs are one of the fastest-growing natural pigment products in the food coloring industry. As microbial fermentation products, MPs are widely used in food coloring, food fermentation [3], and Chinese medicine and food because of their coloring effect, application cost and safety. MPs are secondary metabolites produced by *Monascus* purposes and *Monascus* anka [4] and include yellow, orange, and red azaphilone pigments [5]. In particular, *Monascus* red pigments (MRPs) have promising applications because their red color can be added to various foods. Moreover, MRPs have beneficial effects in controlling blood cholesterol and preventing diabetes, obesity, and cancer [6,7]. MRPs can be added to most products, as a safe food coloring according to production needs [8]. MRPs are widely used in products such as meat, wine, and pasta. In addition to their use in traditional coloring, MRPs are used for product preservation and flavor enhancement. However, MRPs are sensitive to several environmental factors, such as pH, high temperature, oxidation, enzymes, metal ions, and light [9]. Consequently, MRPs exhibit instability, degradation, and discoloration during storage and processing, which limit their practical applications.

Whey proteins and their hydrolysates are valued as essential emulsifiers in foods [10]. Whey protein refers to a group of milk proteins [11]. This protein consists mainly of 50% β-lactoglobulin (molecular weight of approximately 18.3 kDa) and 20% α-lactalbumin (molecular weight of approximately 14.0 kDa) [12]. The molecular structure of whey protein consists of abundant branched-chain amino acids (leucine, isoleucine, and valine), and its total amino acid composition is similar to that of skeletal muscles [13,14,15]. Whey protein products include whey protein isolate (WPI) and whey protein concentrate, which are increasingly used in the food industry due to their high nutritional value. WPI (protein content ≥ 90%) mainly includes α-lactalbumin (α-La), β-lactoglobulin (β-Lg), bovine serum albumin (BSA), immunoglobulin (Ig), and glycopeptides [16]. WPI has gained attention due to its outstanding properties, such as stability, gelling ability, emulsification, nutritional properties, accessibility, nontoxicity, and low cost. Whey protein hydrolysate (WPH) exhibits various biological activities associated with improved pigment stability [17]. Due to its stability, WPI is an essential ingredient and is widely used to improve the textural properties of pigments [18]. Furthermore, owing to its enhanced functional properties, WPI can meet higher food industry requirements [19]. Using WPI and WPH as binding agents to MRPs may enable the development of complexes with excellent stability.

In fact, MRPs are unstable in adverse environments, such as nonideal temperature, light, and pH; moreover, they exhibit relatively low color retention as compared to artificial colors. To overcome these problems and increase the utility of MRPs in the food industry, we combined MRPs with WPI and WPH to form complexes under different pretreatment conditions and assessed the stability of these complexes. However, there are only a few reports on the interactions between MRPs and proteins. Therefore, this study aimed to investigate the preparation, multispectral characterization, and stability of MRPs and WPI complexes, including their binding constant, binding capacity, and surface hydrophobicities. Fluorescence spectroscopy, ultraviolet–visible (UV–Vis) absorption, Fourier transform infrared (FTIR) spectroscopy, and scanning electron microscopy (SEM) techniques were applied. The results of this study provide a theoretical basis for further research on the interaction between MRPs and WPI.

## 2. Materials and Methods

### 2.1. Materials

WPI (protein content > 90%) was procured from Hilmar Ingredients Co. (Hilmar, CA, USA). MRPs were purchased from Henan Qianzhi Trading Co., Ltd. (Zhengzhou, China). Alcalase 2.4 L was purchased from Novozymes Biotechnology Co., Ltd. (Copenhagen, Denmark). The 1-(anilino) naphthalene-8-sulfonate (ANS) reagent was obtained from Sigma (SL, St. Louis, MO, USA). All the other chemicals used in this study were of analytical-grade purity or higher.

### 2.2. Preparation of WPH

Ultrapure water was used to suspend WPI at a concentration of 0.1 g/L, which was hydrated under constant stirring at 25 °C for 2 h. Then, the mixture was refrigerated at 4 °C overnight to facilitate dissolution. To obtain WPI hydrolysates with varying degrees of hydrolysis, Alcalase 2.4 L (GE enzyme) was added at four different concentrations to the WPI: 0.000%, 0.125%, 0.250%, and 0.348%. Next, the mixture was subjected to hydrolysis in a water bath at 40 °C for 4 h. The reaction was quenched by heating the suspension in a water bath at 55 °C for 30 min to inactivate the Alcalase. After the reaction, WPI solids with hydrolysis degrees (DH) of 0.00%, 0.50%, 1.00%, and 1.38% were obtained using lyophilization in a Lyophilizer (LGJ-12F, Foring Technology Development (Beijing) Co., Ltd., Beijing, China).

The DH of WPH was determined using the pH-STAT method [20]. NaOH (0.05 M) was added to keep the reaction system at pH 7.0 [21], and the consumed volume of that was recorded. The DH was calculated using the following equation:(1)$$\mathrm{DH}(\%)=\frac{B\times N_{b}}{\alpha \times M_{p}\times h_{tot}}\times 100$$
where, *B* is the amount of NaOH consumed (mL), *N*_b_ is the concentration of NaOH (M), *α* is the average degree of dissociation of the α-NH_2_ group (pH 7.0, 40 °C, *α* = 0.99), *M*_p_ is the weight of the sample (g), and *h_tot_* is the total number of peptide bonds in WPI (8.8 M) [22].

### 2.3. Surface Hydrophobicity

The surface hydrophobicity was evaluated using 1-anilinonaphthalene-8-sulphonate (ANS) as a fluorescent probe, following the method described by Li et al. [23], but with minor modifications. First, 2.5 mM ANS solutions were prepared and their fluorescence intensities were measured using the F-7000 fluorescence spectrophotometer (Hitachi, Japan). Then, ANS solutions at different concentrations (5, 6.67, 10, 20, and 100 μM) were added to WPI and WPH, and their fluorescence intensities were measured. The measurement parameters were as follows: an excitation wavelength of 390 nm, an emission wavelength of 470 nm, and a constant excitation and emission slit of 5 nm. The surface hydrophobicity was calculated using the following Lineweaver–Burk equation:(2)$$\frac{1}{F}=\frac{1}{F_{\max}}+\frac{K_{d}}{L_{0}}\times \frac{1}{F_{\max}}$$
where, *L*_0_ is the concentration of the ANS fluorescent probe (μM); *F_max_* is the maximum relative fluorescence intensity when the probe concentration is saturated; *K_d_* is the dissociation constant of ANS and the protein-forming complex; and *F_max_/K_d_* (protein concentration) is the surface hydrophobicity index of the protein, i.e., PSH.

### 2.4. Maximum Binding Constants

Fluorescence quenching was utilized to study the binding behavior between WPI and MRPs. Protein solutions with different DHs were prepared at a concentration of 10 g/L. MRPs of 0, 0.2, 1.0, 2.0, 6.0, and 8.0 U (color value unit) were added to these solutions, shaken, and mixed. Then, the samples were incubated in a water bath at 40 °C for 4 h, and the built-in fluorescence spectra were subsequently measured. The binding constants of the complexes were determined via fluorescence spectrometry using distilled water as the blank. The measurement parameters were as follows: an excitation wavelength of 285 nm, an emission wavelength of 300–420 nm, and an excitation and emission slit of width 5 nm. The binding constants were calculated using the static quenching equation. Assuming that the smaller molecule binds independently to a set of equivalent sites on the macromolecule, the binding constant *K_A_* and the number of binding sites *n* can be obtained using a double logarithmic regression curve of lg[(*F*_0_ − *F*)/*F*] versus lg[Q] according to Equation [24]:(3)$$\mathrm{lg}(\frac{F_{0}-F}{F})=\mathrm{lg}K_{A}+n\mathrm{lg}[Q]$$
where, *F*_0_ is the fluorescence intensity of the protein at baseline, *F* is the fluorescence intensity after the addition of MRPs, [*Q*] is the color value (U) of the added MRPs, *K_A_* is the binding constant (U^−1^), and *n* is the number of MRPs bound per WPI.

### 2.5. Maximum Binding Capacity

MRPs (0, 20, 40, 60, 80, 100, 120, 140, 160, 180, 200, 220, and 240 U/g) were added to the 2.5 g/L WPI solution with the highest binding constant, respectively, and were shaken, mixed, and incubated in a water bath (HH-6, Guohua Electric Co., Changzhou, China) at 40 °C for 4 h. The absorbance values at 500 nm were measured before and after incubating the samples in the water bath. The maximum binding capacity of WPI with MRPs was reached when the difference in absorbance before and after heating the samples solution no longer increased substantially [25].

### 2.6. Sodium Dodecyl Sulfate–Polyacrylamide Gel Electrophoresis (SDS–PAGE)

WPI solutions with different DH were diluted to 2 mg/mL using distilled water. Then, the diluted protein solution (40 μL) was taken and a buffer (10 μL) containing 1% SDS and 2% 2-mercaptoethanol was added and centrifuged at 1000 rpm for 1 min and boiled for 5 min [25]. The sample (10 μL) was taken for loading. The voltage was 60 V for the concentrated gel (5%) and 120 V for the isolated gel (12%). Finally, the gel was dyed using 0.25% Coomassie Brilliant Blue R250 solution, and decolorized solution using a mixture solution which contained 12.5% methanol and 10% glacial acetic acid. Gels were scanned using Gel Image Analysis System (MINI Space 1000, Tanon Science & Technology Co., Ltd., Shanghai, China)

### 2.7. Structural Characterization of the Complexes

#### 2.7.1. Preparation of the Complexes

The MRPs solution with the maximum binding capacity were added to the WPI solution with maximum binding constants, mixed, and incubated in a water bath at 40 °C for 4 h. Finally, the samples were freeze-dried to obtain solid complexes and solid WPH [25].

#### 2.7.2. Fourier Transform Infrared (FTIR) Spectroscopy

The samples were ground, filtered through a sieve (D < 0.149 mm), and analyzed using a FTIR spectrophotometer (NEXUS670, Thermo Nicolet Co., Ltd., MDN, Ramsey, MN, USA). The spectra were recorded in the range of 400–4000 cm^−1^, at a 4 cm^−1^ resolution, and using 16 scans. The measurements were performed in a dry environment (25 ± 0.5 °C). One gram of the sample was used for each analysis [26].

#### 2.7.3. UV–Vis Absorption Spectrum

The complexes and WPH at a 5 mg/mL concentration were scanned using UV–Vis light (Evolution 220, Thermo Fisher Scientific, WLM, Waltham, MA, USA). Scanning was performed in the range of 200–700 nm. The initial spectra were recorded using the following scanning parameters: a slit width of 1 nm, a scanning speed of 60 nm/min, and a spectrum step of 1 nm. Moreover, 4 mL quartz cuvettes were used; the thickness of the optical layer was 1 cm [27].

#### 2.7.4. Circular Dichroism (CD) Spectrum

Circular dichroism (CD) spectrum is employed to detect secondary structural modifications of proteins in aqueous solutions that interact with a small molecule as a ligand [24]. The complexes and WPH (concentration = 0.25 mg/mL) were analyzed via CD spectroscopy (J-1500, JASCO Co., Ltd., Tokyo, Japan). The scan range and scan rate were 190–280 nm and 100 nm/min, respectively [28].

#### 2.7.5. Scanning Electron Microscopy (SEM)

Surface ultrastructural observations of the solid complexes and solid WPH were obtained by using SEM (S-3000N, Hitachi Co., Ltd., Tokyo, Japan) [29]. For better sample morphology observation, the SEM was operated with an acceleration voltage of 15.0 kV and the working distance was adjusted to 23.2 mm. The fine surface morphology and structural features of the samples were observed with 100 times magnification.

### 2.8. Stability Evaluation and Mechanism

The enzymatically hydrophobically modified complex (DH = 0.5%) was obtained by adding MRPs to the WPI and WPH solutions with the largest binding constant, mixing thoroughly in a constant-temperature water bath at 40 °C, and incubating for 4 h. The following analyses were performed sequentially (MRPs was the blank group and complex (DH = 0.0%) was the control group). The stability was determined by calculating the color retention rate (A/A_0_) [30], where A_0_ and A were the absorbance before and after treatment at 500 nm, respectively.

#### 2.8.1. Thermal Stability

Complex (DH = 0.0%), complex (DH = 0.5%), and MRPs solution with the same concentration were placed in 50 °C, 60 °C, 70 °C, 80 °C, 90 °C, and 100 °C water baths for 5 h in the dark. The absorbance of the samples was measured hourly.

In food processing and production, the food quality index is typically modeled using either a zero- or first-order kinetics model. The first-order dynamic model is the most widely used. Based on the analytical data on the stability of MRPs in water baths at different temperatures, it was concluded that a first-order kinetic model was appropriate for describing the degradation of MRPs. In the first-order kinetic model, the half-life (t_1/2_) values were determined by plotting the natural logarithms of the A/A0 ratios against the storage times in days, and heating or digestion time in hours [31]:(4)$$\mathrm{lg}(\frac{A}{A_{0}})=-k_{1}t$$
(5)$$t\frac{1}{2}=\frac{\ln2}{k_{1}}$$
where, *A /A*_0_ is the color retention rate, *t* is the processing time, and *−k*_1_ is the slope of the linear regression curve yielded.

#### 2.8.2. Light Stability

The light stability was evaluated using the method described by Long et al. [32] but with small modifications. The same concentrations of complex (DH = 0.0%), complex (DH = 0.5%), and MRPs solution were irradiated (500 l× light intensity, and 30 cm irradiation distance) for 0 to 5 h at room temperature (with temperature adjusted to 25 °C) with an incandescent lamp. The absorbance of the samples was measured hourly.

#### 2.8.3. pH Stability

To compare the pH stability, the pH of complex (DH = 0.0%), complex (DH = 0.5%), and MRPs solution with the same concentration was adjusted to 2.0, 3.0, 4.0, 5.0, 6.0, 7.0, 8.0, 9.0, and 10.0 using 0.1 M NaOH or HCl solution and stored in the dark for 5 h at room temperature [32].

#### 2.8.4. Storage Stability

Complex (DH = 0.0%), complex (DH = 0.5%), and MRPs solutions of the same concentration were placed in a refrigerator at 4 °C for 0, 2, 4, 6, 8, and 10 d. To avoid the effect of light (i.e., to produce “dark” conditions [33]), the vials containing the samples were wrapped in aluminum foil. The absorbance was measured every 2 d.

#### 2.8.5. Oxidant and Reducing Agent Stability

Complex (DH = 0.0%), complex (DH = 0.5%), and MRPs solution with the same concentration were added to separate test tubes. Then, 0.0, 0.2, 0.4, 0.6, and 0.8 M of H_2_O_2_ solution and 0.00, 0.06, 0.12, 0.18, and 0.24 mg/mL of Na_2_SO_3_ solution were added [34]. Finally, they were incubated for 1 h in the absence of light and their absorbance was measured.

#### 2.8.6. Stability under Simulated Gastric and Intestinal Fluid Conditions

Pepsin (2.5 g) was dissolved in 225 mL of distilled water, and the pH was adjusted to approximately 1.0 using HCl solution (0.1 M). Then, sodium chloride (0.5 g) was added to distilled water and the volume was fixed to 250 mL. Complex (DH = 0.0%), complex (DH = 0.5%), and MRPs solution with the same concentration were placed in conical flasks containing the same amount of artificial gastric juice. In the flasks, the shaker parameters were set at 37 °C and the speed was 100 rpm (simulating gastrointestinal peristalsis) [35]. Samples were taken every hour and the absorbance was measured.

Potassium dihydrogen phosphate (1.7 g) and trypsin (2.5 g) were dissolved in distilled water and the volume was fixed to 250 mL. The pH was adjusted to 6.8 using 0.1 M NaOH solution. Complex (DH = 0.0%), complex (DH = 0.5%), and MRPs solution with the same concentration were placed in a conical flask containing the same amount of artificial intestinal fluid. The definitive flask was maintained at a temperature of 37 °C and a shaking speed of 100 rpm (to simulate gastrointestinal peristalsis). Absorbance was measured at 1 h intervals (zero for blank intestinal fluid) [35].

### 2.9. Statistical Analysis

All experiments were performed in triplicate, and the data were reported as the mean ± standard deviation (SD). Statistical analysis was performed using the software Statistical 8.0 (StatSoft Inc., Tulsa, OK, USA). Independent samples t-test and one-way analysis of variance with Tukey’s post hoc test at 95% probability were used to identify the differences among different groups. *p* < 0.05 was regarded as statistically significant.

## 3. Results and Discussion

### 3.1. Surface Hydrophobicity

The surface hydrophobicity can reflect the exposure of hydrophobic groups and the formation of hydrophobic regions in proteins. The hydrophobicity of WPI gradually increased with the degree of GE enzyme hydrolysis (Table 1). The surface hydrophobicity of WPI increased consistently under different degrees of hydrolysis, increasing to 2592.18 at a DH of 1.38%. WPI is a globular protein with hydrophilic regions on its surface and hydrophobic regions within the molecule. After enzymatic hydrolysis, the internal peptide chain of WPI is unfolded and more hydrophobic groups are exposed, thereby increasing the hydrophobicity of the WPI surface. Farup et al. [36] found that hydrolysis products generally have a higher diffusion rate than natural proteins and exhibit improved binding to other substances.

### 3.2. Maximum Binding Constants

As shown in Figure 1a, the relative fluorescence intensity of the solution decreased with increasing MRP level. Furthermore, the graph in Figure 1a shows that the amount of MRPs added positively correlated with the relative fluorescence intensity of the solution and had a good linear relationship (R^2^ = 0.99). The binding constants of WPI and MRPs were calculated using Equation (3), as shown in Table 2. With increasing DH, the binding constant *K_A_* of the MRPs first increased and then decreased, and the largest binding constant was found for WPI (DH = 0.50%) at 0.670 ± 0.06 U^−1^. Thus, enzymatic hydrolysis favored an increased binding capacity of WPI to MRPs, which was strongest when the DH was 0.50%. One possible reason for this result is that the enzymatic hydrolysis of the WPI peptide chain produced a number of small molecule peptides that were more readily bound to MRPs. Panyam et al. [37] reported that proteolysis can proceed either sequentially, releasing one peptide at a time, or through the formation of intermediates that are further hydrolyzed to smaller peptides as proteolysis progresses, which is often termed the ‘zipper mechanism’. Another possible reason is the hydrophobic modification of the WPI by the GE enzyme, where the exposed hydrophobic groups are linked to the non-polar groups in the pigment molecule by hydrophobic interactions, also resulting in a higher binding constant for the WPI to the MRPs. The number of binding sites between the MRP and the WPI is approximately one, indicating that one WPI molecule binds to one MRP molecule in their interaction to form a complex.

### 3.3. Maximum Binding Capacity

The binding constant of MRPs was the highest when the hydrolysis of WPI was 0.50%, and hence WPI (DH = 0.50%) was chosen to study the maximum binding capacity with MRPs. As shown in Figure 1b, the absorbance difference at 500 nm before and after binding became larger as the amount of MRPs added increased. When the absorbance difference of the sample no longer increases, the binding capacity of MRPs to WPI reaches a maximum (180 U/g). As MRPs are gradually added to WPI, their binding sites change from an unsaturated state to a saturated state because of the equilibrium between the MRPs and WPI. When all binding sites were bound to MRPs, they reached a supersaturated state [25]. When MRPs were continued to be added, the amount of WPI bound to MRPs did not change significantly. Therefore, the results demonstrated that free MRP molecules were in dynamic equilibrium with bound MRP molecules.

### 3.4. SDS–PAGE

Figure 2 shows SDS–PAGE plots of WPI and WPH with three different molecular weight protein fractions (band 1: 15 kD, band 2: 18 kD, and band 3: 25 kD) isolated from all samples. The protein bands were small and narrow, possibly due to the small sample size, low protein concentration, and presence of protein impurities. The most abundant band in the WPI is 1 (15 kD) and the least abundant band is band 3 (25 kD). WPI solutions with hydrolysis degrees of 0.50%, 1.00%, 1.50%, and 3.00% were observed. Protein band 2, with DH of 1.50% WPH, became wider and increased in content. These changes may be due to GE enzyme hydrolysis, mainly due to the molecular components of proteins with a molecular weight of 25 kD, while also producing small amounts of other molecular weight proteins. This suggests that Alcalase has a hydrolytic effect on the WPI protein. Comparison of the electrophoretic profiles of the WPI hydrolysates showed a different trend in the brightness of band 2 with increasing DH, indicating that more of the protein components (25 kD) were hydrolyzed to proteins of other molecular weights. Comparing the protein bands with molecular weights between 25 kD and 27 kD revealed that the bands for protein components in this molecular weight range were the brightest and most abundant when DH = 1.50%. Thus, the highest molecular weight of WPI solutions of this magnitude is 18–27 kD, and proteins of this molecular weight may have specific properties. A previous study [38] reported acid-assisted salt-soluble proteins from chickens.

### 3.5. Structural Characterization of WPI (DH = 0.5%) and Complex (DH = 0.5%)

#### 3.5.1. FTIR Spectroscopy

The elements of the secondary structure of the protein were analyzed using FTIR spectroscopy in the spectral range of WPI and complex (DH = 0.5%). FTIR analysis revealed the presence of several absorbance peaks related to some specific functional groups (Figure 3a), including stronger characteristic peaks of WPI and complex (DH = 0.5%) at 1661.32 cm^−1^ corresponding to the stretching vibration of the carbonyl group (C=O), as well as a peak at 1249.65 cm^−1^ corresponding to the stretching vibration of C–O, and a peak at 3070.12 cm^−1^ corresponding to the stretching vibration of C–H. The stronger characteristic peaks of WPI and complex (DH = 0.5%) at 1643.05 cm^−1^ is caused by the carbon–carbon double bond stretching vibrations. The peak near 1560 cm^−1^ and 3300 cm^−1^ are caused by stretching vibrations of the aromatic ring C=C double bond. The distinct peaks near 3400 cm^−1^ are caused by the stretching vibrations of N–H single bonds, such as peptide bonds. The absorption peaks across the spectrum become significantly weaker after adding MRPs, indicating that the combination of WPI and MRPs changes the peptide bond, single hydrocarbon bond, carbon–oxygen double bond, and carbon–carbon double bond, and that the stretching vibrations are significantly weaker. Nevertheless, the perturbation was related to various interactions, including hydrogen bond, covalent bonds and van der Waals forces; moreover, these forces could compete with intermolecular forces among amino acid residues [39]. Additionally, the average peak was found to be slightly shifted toward the higher wavenumbers. Using FTIR spectroscopy, a previous study [40] found that the most pronounced and multidirectional changes in the secondary structure of the protein occurred under the influence of the inactive form of cortisol, with results similar to this present experiment.

#### 3.5.2. UV–Vis Spectroscopy

As shown in Figure 3b, the UV–Vis absorption spectra of WPI and the complex also differed due to the binding of WPI to MRPs. The UV–Vis absorption peaks of the WPI hydrolysate appeared at 225 and 264 nm due to the characteristic absorption peaks of the WPI peptide bond near 225 nm and the phenylalanine residue near 260 nm, respectively. The UV–Vis absorption spectrum of the composite solution showed five absorption peaks at 206, 232, 273, 408, and 488 nm compared with the absorption spectrum of WPI alone. A slight redshift in the characteristic absorption peaks of the peptide bond and phenylalanine was observed, and the whole spectrum also showed significant color enhancement. First, the peak at 206 nm is probably due to the enhanced conjugation among the unsaturated carbonyl group of the damiana red pigment, aromatic amino acid residues, and the backbone of WPI. The peaks at 232 nm and 273 nm are the absorption peaks of the peptide bond and phenylalanine residues of the protein, respectively. The peaks at 408 and 488 nm are characteristic absorption peaks for the red pigment of turbot. The UV–Vis absorption of the system has absorption peaks at 408 and 488 nm due to the presence of MRPs in the complex. The interaction between the MRPs and WPI enhanced conjugation and increased the chromophores, redshift, and color enhancement. Static quenching of a substance in combination with a quenching agent usually changes the UV–Vis absorption spectrum of the substance. The change in the UV–Vis spectrum of WPI after adding MRP solutions indicates that the quenching effect of MRPs on WPI is mainly static quenching, suggesting that WPI and MRPs interact with each other in the system [1].

#### 3.5.3. CD Spectroscopy

Proteins, such as macromolecules with photoactive groups, can produce CD. CD spectroscopy can be used to quickly analyze the conformational changes of the protein caused by ligand addition [41]. As shown in Figure 3c, CD spectroscopy showed one positive peak at 192 nm and two negative peaks at 208 nm and 218 nm, typically characteristic of an α-helix structure. In contrast, a β-fold shows a robust negative peak at 216 nm and a positive peak between 195 and 198 nm. Therefore, both WPI and the complex contain β-folded secondary structures. These results suggest that the interaction of WPI with MRPs disrupts the α-helix structure and some random coils in WPI proteins. The molecule reorganizes to form a β-fold, and the CD peak is significantly weakened. A large amount of β-folding in the WPI redshifts the positive peak of the α-helix at 205 nm, and the results suggest some conformational changes in the secondary structure of WPI. Nevertheless, the perturbation was related to interactions, including hydrogen bond or van der Waals forces, and these forces could compete with intermolecular forces among amino acid residues. For example, a change in the secondary structure, i.e., a decrease in the proportion of α-helices, was registered during its interaction with testosterone [42].

#### 3.5.4. Scanning Electron Microscopy (SEM)

SEM is an indispensable tool that uses a high-energy electron beam to excite various information on the surface of a substance through the interaction between the substance and the electron beam, and then analyzes this collected information to characterize the microscopic form of the substance. As shown in Figure 3d, complex (DH = 0.5%) had some rod and block shapes. The SEM image of WPI (DH = 0.5%) had smoother and disordered lamellar structures (Figure 3e). The low content of MRPs (MRPs to WPI mass ratio of 1:44) influences the surface morphology of the WPI hydrolysate. The interaction of a ligand with a protein can be accompanied by conformational changes in the latter. Ali et al. [43] found that when the MP–SC (*Monascus* pigments–sodium caseinate) complex was formed due to the interaction between MPs and SC, its morphology was more stable, and the complex distribution was clear. The results show that pigments binding to proteins can cause conformational changes in proteins.

### 3.6. Stability of Complex (DH = 0.0%), Complex (DH = 0.5%), and MRPs

#### 3.6.1. Temperature Stability

MRPs are natural pigments that decompose when subjected to heat, causing discoloration, especially after high-temperature treatment, with severe loss of pigment [44]. Considering the industrial application of the pigment, the thermal stability of the pigment was measured for 5 h at temperatures between 50 °C and 100 °C (Figure 4). The results showed that the color value retention of complex (DH = 0.0%), complex (DH = 0.5%), and MRP solutions were >80% after 5 h in a water bath at 50 °C. There was no significant difference in stability among the three groups (*p* > 0.05), indicating that this temperature had a negligible effect on the stability of the three groups of samples. When the temperature was >50 °C (Figure 4a), the color value retention rate of MRPs gradually decreased as the temperature increased. When the temperature was 100 °C (Figure 4f), the color value retention rate of MRPs was only approximately 40% in the water bath for 1 h; after 5 h, the color value retention rate of MRPs was approximately 20%. This indicates that high temperature has a significant effect on the stability of MRPs. When the temperature was 100 °C, the color value retention of the complexes (DH = 0.0%) and (DH = 0.5%) were slightly higher than that of MRP solution after 5 h of water bath treatment in the three sample groups; however, there was no significant difference in color value retention between the three groups (*p* > 0.05). At 60 °C–90 °C (Figure 4b–e), the color value preservation of the complex (DH = 0.0%) and complex (DH = 0.5%) were significantly higher than that of the MRPs (*p* < 0.05). This indicates that the complexes formed by WPI and MRPs not only improved the binding ability of MRPs but also improved the temperature stability of MRPs. Adding WPI to the pigments can enhance their solubility under acidic conditions due to the hydrophilic and hydrophobic groups of WPI.

The isothermal model of MRPs is reported to follow first-order reaction kinetics [31]. Table 3 shows the results of kinetic studies on the thermal degradation of both complexes and MRPs in a water bath at 50–100 °C. The linear relationship shows that the thermal degradation of MRPs follows first-order reaction kinetics. The t_1/2_ values of MRPs decreased from 19.64 h to 1.82 h at 50–100 °C, indicating that MRPs are sensitive to high temperatures. Given these results, we observed that the t_1/2_ values of complex (DH = 1.0%) and complex (DH = 0.0%) were more than twice as high as those of MRPs at 60 °C and 70 °C, suggesting that binding to WPI and its hydrolysates significantly improved the thermal stability of MRPs. In bulk, WPI starts to unfold at 50 °C and can aggregate and form intermolecular bonds at 70 °C [45]. Thus, WPI complexes have significant color stability at 60 °C and 70 °C, probably due to the formation of abundant intermolecular bonds between the pigment and the WPI protein, resulting in the protection of the pigment. Furthermore, these results suggest that temperature control of pigments is crucial in the food industry.

#### 3.6.2. Light Stability

As shown in Figure 5a, after 1 h of sunlight irradiation, the color value preservation of complex (DH = 0.0%), complex (DH = 0.5%), and MRP solutions rapidly decreased to <40%. After 5 h of irradiation, the color value preservation rates for the three groups of samples were 15.0%, 14.0%, and 7.8%, respectively. This indicates that sunlight significantly affects the stability of the complexes and MRPs. The retention of complex (DH = 0.0%) and complex (DH = 0.5%) were significantly higher than that of MRPs under the same light conditions (*p* < 0.05), indicating that binding to WPI was effective in improving the color retention of MRPs. Pigmentation likely occurs due to photosensitive reactions of free radicals and oxidative discoloration. A previous study [46] proposed a mechanism for photo-fading. Due to the presence of oxygen, the redox reactions of three free radicals, aliphatic side chain breaks, hydroxyl radicals (OH^−^), and superoxide anions (O_2_^−^) lead to the discoloration of aqueous solutions of MRPs. To improve the stability of natural pigments, pigment stabilizers can be added.

#### 3.6.3. Storage Stability

MRPs are stable at room temperature, so there is a minimal loss of pigment when stored for short periods at 4 °C. However, oxidation changes the color of the MRPs due to the presence of small amounts of oxygen in the solution, resulting in a decrease in the retention rate of the MRPs’ color values. As shown in Figure 5b, the color retention rates of complex (DH = 0.0%), complex (DH = 0.5%), and MRPs gradually decreased as the storage time increased. After 10 d of dark storage, the color value preservation of both complex (DH = 0.0%) and complex (DH = 0.5%) were >95%, whereas that of the MRPs was only 80%. The MRPs were bound to WPI (DH = 0.0%) or WPI (DH = 0.5%) via enzymatic hydrolysis, reducing the chance of interaction with oxygen. Therefore, the retention rates of complex (DH = 0.0%) and complex (DH = 0.5%) were significantly higher than those of MRPs under the same storage conditions (*p* < 0.05), indicating that WPI formed complexes with MRP, which significantly improved the storage stability of MRPs. Long et al. [32] reported that MRPs have lower degradation when treated with storage in the dark, which is outlined with the results of this study.

#### 3.6.4. pH Stability

Since β-Lg and α-Lac are the major fractions of whey protein, conformational changes in these fractions caused by environmental conditions can be used to explain the binding behavior of WPI under different conditions. pH influences the retention of pigments in MRPs. Under strong acidic conditions, solubility decreases, leading to the possible precipitation of MRPs from the solution and a consequent decrease in color. As shown in Figure 5c, within a certain pH range (2–10), the retention of MRP pigments is stable and their color is essentially constant. At pH 2, the color value retention of MRPs was only 70%, which was significantly lower than that of the two complexes. The isoelectric point of WPI is pH 4–5 [37], so when the solubility of the protein reaches the isoelectric point (having a net charge of 0) at pH 4 and pH 5, the protein is prone to aggregation and precipitation, hence, complex (DH = 0.0%) and complex (DH = 0.5%) reach the isoelectric point at pH 4 and pH 5 at 500 nm. The absorbance values at 500 nm were too high (>1), resulting in the accumulation and precipitation of both groups of samples, with color value retention of >1. The results showed that the color value retention of complex (DH = 0.0%) and complex (DH = 0.5%) were significantly higher than that of MRPs under different pH conditions (*p* < 0.05), indicating that the combination with WPI was beneficial for improving the color retention of MRPs.

#### 3.6.5. Stability of Oxidizing and Reducing Agents

Hydrogen peroxide is a strong oxidant bleaching agent. It is generally accepted that hydrogen peroxide bleaching functions by ionizing H_2_O_2_ in water, which then reacts with the pigment to cause a color change. MRPs are stable in low concentrations of H_2_O_2_, but when more amounts are added, some pigment is lost due to the bleaching effect of hydrogen peroxide, resulting in reduced retention of the color value. According to Figure 6a, the color retention rates of complex (DH = 0.0%), complex (DH = 0.5%), and MRPs were all >95% after treatment with low concentrations of H_2_O_2_ (0–0.8 mg/mL), indicating that low concentrations of H_2_O_2_ had little effect on MRP stability. Under the same conditions, the color retention rate of complex (DH = 0.0%) and complex (DH = 0.5%) was higher than that of MRPs. After enzymatic hydrolysis, the combination of MRPs with WPI (DH = 0.0%) or WPI (DH = 0.5%) reduced the probability of interaction with H_2_O_2_. Consequently, the color value retention rate was significantly higher for complex (DH = 0.0%) and complex (DH = 0.5%) than the MRPs after treatment with the same concentration of H_2_O_2_ (*p* < 0.05). These results indicate that the complexes formed by WPI and MRPs improved the stability of MRPs to oxidants. A previous study [47] suggested that sericin-induced melanogenesis in cultured RPE is associated with elevated levels of superoxide dismutase, hydrogen peroxide, and inflammatory proteins.

Na_2_SO_3_ is a strong reducing agent that enhances the color of pigments and can be used as a preservative in the food industry. When coloring food, Na_2_SO_3_ can be added to enhance not only the color but also the preservative ability of food. According to Figure 6b, the color value retention rate of complex (DH = 0.0%), complex (DH = 0.5%), and MRPs were >97% after adding a low concentration of Na_2_SO_3_ (0.00–0.30 mg/mL), indicating that a low concentration of Na_2_SO_3_ had little effect on the stability of MRPs. After treatment with a low concentration of Na_2_SO_3_, the color value retention of complex (DH = 0.0%) and MRPs were >1, indicating that a low concentration of Na_2_SO_3_ had some color protection effect on MRPs. These results showed that the color value retention of the complex (DH = 0.0%) was higher than that of MRPs, and there was a significant difference between the two groups (*p* < 0.05), indicating that the complex formed by WPI and MRPs (DH = 0.0%) not only improved the stability of MRPs to the reducing agent but also enhanced the color value retention of MRPs by Na_2_SO_3_. The color of the pigment was enhanced.

#### 3.6.6. Stability under Artificial Gastric and Intestinal Fluid Conditions

The harsh conditions of the gastrointestinal (GI) environment are a major obstacle affecting pigmentation [48]. In the simulated gastric fluid environment, the solubility of MRPs decreases under strong acidic conditions, and the color of large MRPs may precipitate out of the solution, resulting in a decrease in the retention of the color values of MRPs. As shown in Figure 6c, the color retention of complex (DH = 0.0%), complex (DH = 0.5%), and the MRP solution gradually decreased as the digestion time increased. After 5 h of treatment with simulated gastric juice, the color value preservation rates were 92.41% and 93.85% for complex (DH = 0.0%) and complex (DH = 0.5%), respectively, compared with 81.40% for the MRPs. There was a significant difference (*p* < 0.05) between the color retention rate of the two complexes and MRPs. These results indicate that complexes formed by WPI and MRPs can significantly improve the stability of MRPs in the simulated gastric fluid.

In the simulated intestinal fluid environment, the pH was approximately 6.8. Under neutral conditions, MRPs are relatively stable. Figure 6d shows that the retention of color values of complex (DH = 0.0%), complex (DH = 0.5%), and MRPs after 5 h of treatment with simulated intestinal fluid exceeded 95%, indicating that MRPs were relatively stable in simulated intestinal fluid. In addition, the color retention rates of both complex (DH = 0.0%) and complex (DH = 0.5%) were higher than those of MRPs and were significantly different (*p* < 0.05). This suggests that the complexes formed by WPI and MRPs enhanced the stability of MRPs in the artificial intestinal fluids.

## 4. Conclusions

Low bioavailability of MRPs in unfavorable environments makes it important to improve their stability. Complexes are a promising binding modality with potential applications in the pharmaceutical, cosmetic, and food industries. Multispectral techniques and stability experiments were used to elucidate the protective effect of WPI and WPH on the stability of MRPs. Fluorescence spectroscopy, UV–Vis absorption, FTIR, and CD analysis showed that there was an interaction between WPI and MRPs. The results show that interaction between MRPs and WPI was formed by hydrogen bonds (C–H binding) at peak 1450.21 cm^−1^ (which was contributed by methyl groups). Furthermore, the t_1/2_ values of complex (DH = 0.5%) and complex (DH = 0.0%) were twice as high as those of MRPs at 60 °C and 70 °C, suggesting that the binding of MRPs to WPI and WPH greatly improved the thermal stability of MRPs. In this study, MRPs and WPI are held together by hydrophobic interactions and hydrogen bonds, which can change the conformation of WPI, thus effectively protecting the stability of MRPs. Overall, our findings provide a comprehensive understanding of the interactions between WPI and MRPs, which may be useful for studying the different ways MRPs bind to proteins.

## Figures and Tables

**Figure 1 foods-12-01745-f001:**
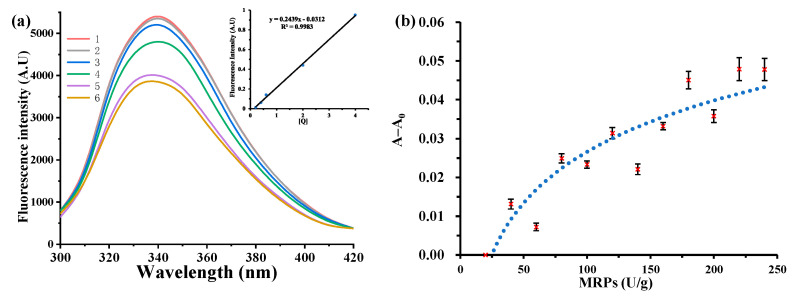
Effect of MRPs on the fluorescence spectra of WPI hydrolysis (**a**), 1–6: 0, 0.2, 1.0, 2.0, 6.0, and 8.0 U MRPs; the binding capacity of WPI (DH = 0.5%) with MRPs (**b**).

**Figure 2 foods-12-01745-f002:**
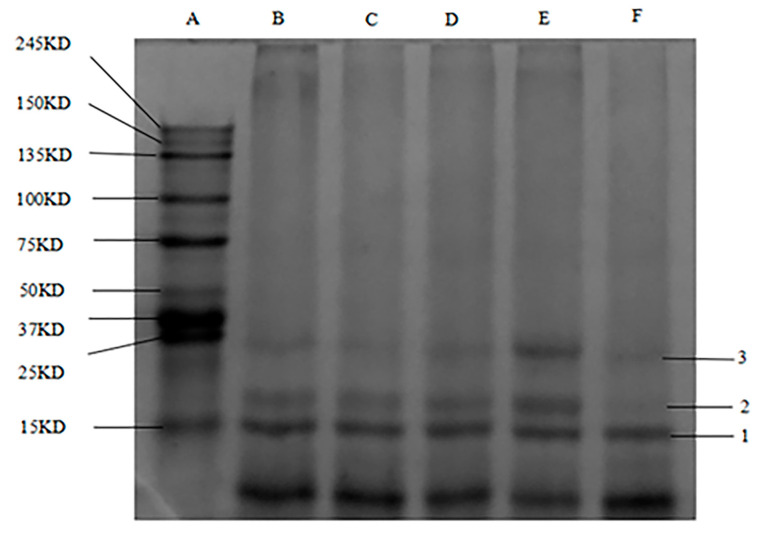
SDS–PAGE profile of WPI solutions with different hydrolysis degrees. A–F: protein mass marker, DH = 0.00%, DH = 0.50%, DH = 1.00%, DH = 1.50%, and DH = 3.00%, respectively. 1–3: 15 kD, 18 kD, and 25 kD, respectively.

**Figure 3 foods-12-01745-f003:**
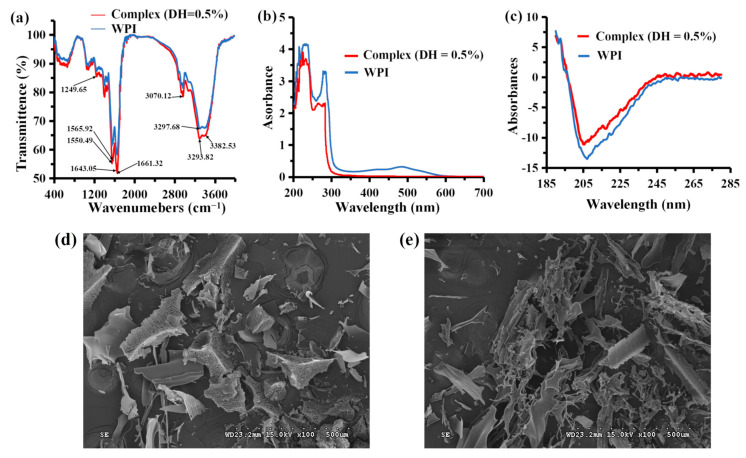
Structural characterization of WPI (DH = 0.5%) and complex (DH = 0.5%) using FTIR spectroscopy. (**a**) UV–Vis spectroscopy; (**b**) CD spectroscopy; (**c**) SEM photograph of complex (DH = 0.5%) (**d**) and WPI (DH = 0.5%) (**e**).

**Figure 4 foods-12-01745-f004:**
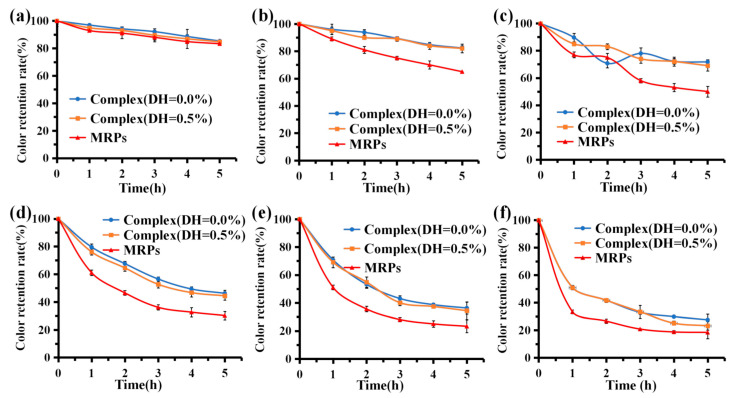
Temperature stability. (**a**–**f**): The color retention rates of complex (DH = 0.5%), complex (DH = 0.0%), and MRPs in a 50 °C, 60 °C, 70 °C, 80 °C, 90 °C, and 100 °C water bath.

**Figure 5 foods-12-01745-f005:**
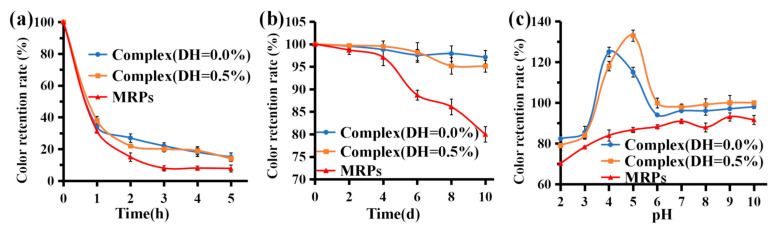
Stability of complex (DH = 0.0%), complex (DH = 0.5%), and MRPs under light conditions. (**a**) 4 °C storage; (**b**) and different pH values (**c**).

**Figure 6 foods-12-01745-f006:**
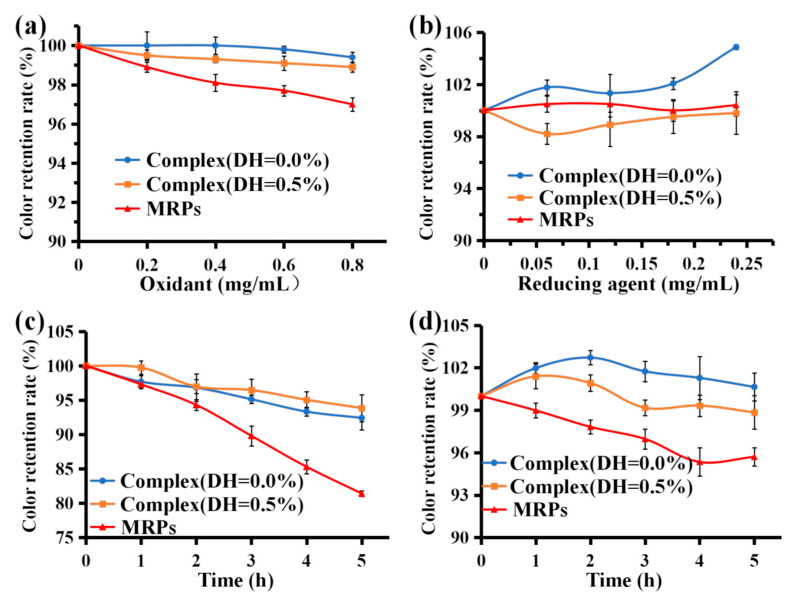
Stability of complex (DH = 0.0%), complex (DH = 0.5%), and MRPs under an oxidant agent (**a**), a reducing agent (**b**), artificial simulated gastric (**c**), and intestinal fluid (**d**).

**Table 1 foods-12-01745-t001:** Surface hydrophobicity of WPI and WPH.

Samples	PSH	R^2^
WPI (DH 0.0%)	2344.03 ± 159.50 ^a^	0.9932
WPI (DH 0.5%)	2370.00 ± 65.04 ^a^	0.9920
WPI (DH 1.0%)	2474.13 ± 234.66 ^a^	0.9931
WPI (DH 1.38%)	2592.18 ± 243.67 ^a^	0.9941

Data with the same letter are not significantly different (*p* < 0.05).

**Table 2 foods-12-01745-t002:** Binding constants of WPI and WPH.

Samples	K_A_/U^−1^	R^2^	*n*
WPI (DH 0.00%)	0.445 ± 0.02 ^b^	0.9938	0.88 ± 0.05
WPI (DH 0.50%)	0.670 ± 0.06 ^a^	0.9821	1.02 ± 0.26
WPI (DH 1.00%)	0.590 ± 0.08 ^a^	0.9902	0.95 ± 0.06
WPI (DH 1.38%)	0.472 ± 0.01 ^b^	0.9870	0.85 ± 0.08

Data in the same column with different superscript letters are significantly different (*p* < 0.05).

**Table 3 foods-12-01745-t003:** Kinetic parameters.

Treatment	Samples	k (h^−1^)	t_1/2_ (h)	R^2^
50 °C	complex (DH = 0.0%)	0.031 ± 0.007 ^a^	22.22 ± 1.68 ^a^	0.9928
	complex (DH = 0.5%)	0.032 ± 0.005 ^a^	21.39 ± 0.90 ^a^	0.9865
	MRPs	0.035 ± 0.018 ^a^	19.64 ± 1.10 ^a^	0.9576
60 °C	complex (DH = 0.0%)	0.040 ± 0.007 ^b^	17.42 ± 0.47 ^a^	0.9901
	complex (DH = 0.5%)	0.039 ± 0.007 ^b^	17.68 ± 0.49 ^a^	0.9772
	MRPs	0.084 ± 0.004 ^a^	8.22 ± 0.21 ^b^	0.9895
70 °C	complex (DH = 0.0%)	0.068 ± 0.006 ^b^	10.18 ± 0.31 ^a^	0.9665
	complex (DH = 0.5%)	0.073 ± 0.009 ^b^	9.50 ± 0.77 ^a^	0.9820
	MRPs	0.147 ± 0.013 ^a^	4.72 ± 0.05 ^b^	0.9862
80 °C	complex (DH = 0.0%)	0.157 ± 0.011 ^b^	4.43 ± 0.16 ^a^	0.9751
	complex (DH = 0.5%)	0.164 ± 0.010 ^b^	4.24 ± 0.23 ^a^	0.9566
	MRPs	0.248 ± 0.007 ^a^	2.79 ± 0.24 ^b^	0.9850
90 °C	complex (DH = 0.0%)	0.199 ± 0.016 ^b^	3.48 ± 0.11 ^a^	0.9822
	complex (DH = 0.5%)	0.211 ± 0.021 ^b^	3.28 ± 0.16 ^a^	0.9722
	MRPs	0.279 ± 0.014 ^a^	2.49 ± 0.16 ^b^	0.9709
100 °C	complex (DH = 0.0%)	0.322 ± 0.035 ^a^	2.15 ± 0.21 ^a^	0.9648
	complex (DH = 0.5%)	0.323 ± 0.024 ^a^	2.14 ± 0.26 ^a^	0.9657
	MRPs	0.380 ± 0.028 ^a^	1.82 ± 0.27 ^a^	0.9543

Data in the same column with different superscript letters are significantly different (*p* < 0.05).

## Data Availability

Data is contained within the article.

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
