# Peer review of "Preparation, Multispectroscopic Characterization, and Stability Analysis of Monascus Red Pigments—Whey Protein Isolate Complex"

_foods, 2023, doi:10.3390/foods12091745_

Round 1

Reviewer 1 Report

The MS is interesting but requires revision.

1. Novelty of the MS should be mentioned in the introduction.

2. More recent and relevant research should be included in the introduction section.

3. Conclusion should be supported with numerical results.

4. Sections 2.3-2.8.5 should be supported with appropriate citations. Previously published research articles should be cited against each of the methods. e.g. the following article may be cited against 2.7.2 Fourier transform infrared (FTIR) spectroscopy

https://doi.org/10.1007/s12010-021-03669-8

SEM study may be cited with the following article https://doi.org/10.1016/j.micron.2007.06.010

5. The following section should be rewritten in detail

2.7.2; 2.7.5; 2.8.4 

6. 3.5.4:SEM analysis should be rewritten in detail

7. English language editing is suggested

Author Response

Response to the first reviewer's questions

  1. Novelty of the MS should be mentioned in the introduction.

Answer: Thank you for your helpful suggestion very much. We have already mentioned the novelty of MS in the introduction with marked by green color (Line 30-47).

  1. More recent and relevant research should be included in the introduction section.

Answer: Thank you for your helpful suggestion very much. We have added more recent and relevant research in the introduction section with marked by green color.

  1. Conclusion should be supported with numerical results.

Answer: Thank you for your helpful suggestion very much. We have added the numerical results to the conclusion with marked by green color (Line 542-550).

  1. Sections 2.3-2.8.5 should be supported with appropriate citations. Previously published research articles should be cited against each of the methods. e.g. the following article may be cited against 2.7.2 Fourier transform infrared (FTIR) spectroscopy https://doi.org/10.1007/s12010-021-03669-8

SEM study may be cited with the following article https://doi.org/10.1016/j.micron.2007.06.010

Answer: Thank you for your helpful suggestion very much. We have added appropriate citation in sections 2.3 2.8.5 with marked by green color.

  1. The following section should be rewritten in detail

2.7.2; 2.7.5; 2.8.4 

Answer: Thank you for your helpful suggestion very much. We have rewritten sections 2.7.2, 2.7.5, and 2.8.4 in detail with marked by green color.

  1. 3.5.4:SEM analysis should be rewritten in detail

Answer: Thank you for your helpful suggestion very much. We have rewritten the 3.5.4 SEM section of the analysis in detail with marked by green color (Line 382-393).

  1. English language editing is suggested

Answer: Sorry for our unclarity expression. We have tried our best to improve the grammar and readability, and corrected some basic errors. Our manuscript has been proofread by a professional editing service, and the certification of English Editing has been sent to Editor. We hope that the current revised manuscript meets your requirements

Author Response

Response to the second reviewer's questions

1、Title should be modified to "Preparation, ..... of Monascus red pigments - whey protein isolate complex"

Answer: Thank you for your professional suggestion. We have modified the article title marked by blue color (Line 3-4).

  1. Replace the WPI hydrolysis product directly with Whey protein isolate hydrolysate (WPH)

Answer: Thank you for your professional suggestion. We have replaced the WPI hydrolysis product directly with Whey protein isolate hydrolysate (WPH) with marked by blue color.

  1. Please provide the SDS-PAGE profile of WPI-MRPs and WPH-MRPs complexes

Answer: Thank you for your professional suggestion. In fact, We tested the SDSPAGE of WPIMRPs and WPHMRPs complexes (Figure 1). However, we found that the SDS-PAGE profile of the WPI-MRP and WPH-MRP complexes were similar to WPI and WPH without significant differences, indicating that the binding of MRPs to WPI and WPH did not alter the distribution of the proteins on the electropherograms, so the SDS-PAGE profile of the WPI-MRPs and WPH-MRPs complexes were not included in the manuscript.

Figure 1. SDS–PAGE profile of WPIMRPs and WPHMRPs complexes with different hydrolysis degrees. A–F: protein mass marker, DH = 0.00%, DH = 0.50%, DH = 1.00%, DH = 1.50%, and DH = 3.00%, respectively.

  1. What about covalent bonds? the Maillard reaction could occur at 40℃ for 4 h

Answer: Thank you for your professional question. We should consider the  Maillard reaction occur at 40℃ for 4 h. Therefore,we have added covalent bonds to the discussion marked by blue color (Line 332-335).

  1. The mechanisms relating to the stability of the MRPs as a result of interaction with WPI should be provided by the authors.

Answer: Thank you for your suggestion. In this study, MRPs and WPI are held together by hydrophobic interactions and hydrogen bonds. Adding WPI to the pigments can enhance their solubility and promote binding under acidic conditions due to the hydrophilic and hydrophobic groups of WPI. We have provided the mechanisms relating to the stability of the MRPs marked by blue color (Line 416-417).

Reviewer 3 Report

The study of Lai et al. is of some interest, but because of the poor language some parts of the paper are difficult to follow. Unfortunately, the language affects significantly the scientific values of the paper.

Some specific comments are the following:

The tile of the study is not relevant. It should be reconsidered. It should indicate that the study deals with the WPI–MRP complexes

The Abstract is not prepared in a coherent manner.

Lines 112, 128: What is the meaning of U MRP?

Line 114: The authors should consider replacing the term “binding” with “incubating” in “…measured after binding in a water bath …” Please check the entire manuscript. The same observation applies to line 129.

Line 127: At line 85 DH was defined as “a hydrolysis degree”. So, how can someone select the hydrolysis degree with the highest binding constant? The language is really poor.

Lines 126-132: It is not clear how the maximum binding capacity was assessed.

Line 135: What is the “sample buffer”?

The authors should avoid starting the sentence with a number.

Line 228: The following statement is not correct: “Surface hydrophobicity is important for assessing the spatial structure of a protein”

The codifications of the samples should be reconsidered.

What is the UM for the surface hydrophobicity?

The terminology used is not always appropriate. The authors failed to correctly explain the main phenomenon standing behind the results obtained. The discussion of the results against literature is poor.

The experimental results presented in tables are provided together with the results of the statistical analysis. Anyway, no indication on how to read the statistical analysis results is mentioned in the footer of the tables.

Author Response

Response to the third reviewer's questions

  1. The study of Lai et al. is of some interest, but because of the poor language some parts of the paper are difficult to follow. Unfortunately, the language affects significantly the scientific values of the paper.

Answer: Sorry for our unclarity expression. We have tried our best to improve the grammar and readability, and corrected some basic errors. Our manuscript has been proofread by a professional editing service, and the certification of English Editing has been sent to Editor. We hope that the current revised manuscript meets your requirements.

  1. The title of the study is not relevant. It should be reconsidered. It should indicate that the study deals with the WPI–MRP complexes

Answer: Thank you for your helpful suggestion very much and sorry for our carelessness, we have revised the title with marked by blue color (Line 3-4).

  1. The Abstract is not prepared in a coherent manner.

Answer: Thank you for your helpful suggestion very much. We have made the summary coherent written with marked by orange color (Line 15-26). And our manuscript has been proofread by a professional editing service, and the certification of English Editing has been sent to Editor.

  1. Lines 112, 128: What is the meaning of U MRP?

Answer: Thank you very much for your professional question. U is the color value unit of MRPs and is one of the main quality indicators of natural pigments [1-3] (Line 120).

  1. Line 114: The authors should consider replacing the term “binding” with “incubating” in “…measured after binding in a water bath …” Please check the entire manuscript. The same observation applies to line 129.

Answer: Thank you for your helpful suggestion very much. We have replaced “binding” with “incubating” in lines 122 and 185 with marked by orange color.

  1. Line 127: At line 85 DH was defined as “a hydrolysis degree”. So, how can someone select the hydrolysis degree with the highest binding constant? The language is really poor.

Answer: Sorry for our unclarity expression. We have tried our best to improve the grammar and readability, and corrected some basic errors. Our manuscript has been proofread by a professional editing service, and the certification of English Editing has been sent to Editor. First, we calculated the binding constants for different degrees of hydrolysis WPI and MRPs, and then select the WPI hydrolysis degree with the highest binding constant.

  1. Lines 126-132: It is not clear how the maximum binding capacity was assessed

Answer: The maximum binding capacity of WPI to MRPs is reached when the absorbance difference no longer increases substantially [4] (139-141).

  1. Line 135: What is the “sample buffer”?

Answer: Sorry for our unclarity expression. Sample buffer means buffer which contains 1% SDS and 2% 2mercaptoethanol. We have modified it with marked by orange color (Line 144).

  1. The authors should avoid starting the sentence with a number.

Answer: Thank you for your helpful suggestion very much. We have revised the sentences that start with the number in the whole manuscript and marked by orange color.

  1. Line 228: The following statement is not correct: “Surface hydrophobicity is important for assessing the spatial structure of a protein”.

Answer: Sorry for our not correct expression. We have removed this sentence.

  1. The codifications of the samples should be reconsidered.

Answer: Thank you for your helpful suggestion very much. We have reconsidered the compilation of the sample.

  1. What is the UM for the surface hydrophobicity?

Answer: Thank you for your professional question very much. But UM does not appear in this study. Protein surface hydrophobicity refers to the surface charge distribution of a protein in aqueous solution. U-1 is the unit of the binding constant marked by orange color [4] (Line 134).

  1. The terminology used is not always appropriate. The authors failed to correctly explain the main phenomenon standing behind the results obtained. The discussion of the results against literature is poor.

Answer: Sorry for our not appropriate terminology. We have revised parts of the discussion and added appropriate literature it in detail with marked by green color.

  1. The experimental results presented in tables are provided together with the results of the statistical analysis. Anyway, no indication on how to read the statistical analysis results is mentioned in the footer of the tables.

Answer: We apologize for not mentioning how to read the results of the statistical analysis in the footer of the table. We have added the statistical analysis with marked by orange color (Line 260, 277 and 435).

References

  1. Bai, J.; Gong, Z.; Shu, M.; Zhao, H.; Ye, F.; Tang, C.; Zhang, S.; Zhou, B.; Lu, D.; Zhou, X.; Lin, Q.; Liu, J. Increased Water-Soluble Yellow Monascus Pigment Productivity via Dual Mutagenesis and Submerged Repeated-Batch Fermentation of Monascus purpureus. Front Microbiol. 2022, 13, 914828. https://doi.org/10.3389/fmicb.2022.914828.
  2. Moghadam, H. D.; Tabatabaee Yazdi, F.; Shahidi, F.; Sarabi-Jamab, M.; Es'haghi, Z. Co-culture of Monascus purpureus with Saccharomyces cerevisiae: A strategy for pigments increment and citrinin reduction. Biocatalysis and Agricultural Biotechnology. 2022, 45, 102501. https://doi.org/10.1016/j.bcab.2022.102501.
  3. Hu, Y.; Zheng, Y.; Liu, B.; Gong, Y.; Shao, Y. Mrhst4 gene, coding for NAD+-dependent deacetylase is involved in citrinin production of Monascus ruber. J Appl Microbiol. 2023, 134. https://doi.org/10.1093/jambio/lxad042.
  4. Xiaoquan, Y.; Jiangyin, Y.; Junjie, H. Effect of Enzymatic Modification on Binding Capacity between Pigment and Glycinin. Modern Food Science and Technology. 2013, 29, 2566-2571. https://doi.org/10.13982/j.mfst.1673-9078.2013.11.015.

Reviewer 4 Report

Interesting paper. The Authors used many methods to make full characterization of obtained complexes.

English language level is satisfactory.

Change the title on:

Preparation, multispectroscopic characterization, and stability analysis of Monascus red pigments and whey protein isolate complex

One important paper is missing and it should be cited in the Discussion.

YANG Xiao-quan, YAN Jiang-yin, HOU Jun-jie. Effect of Enzymatic Modification on Binding Capacity between Pigment and Glycinin. Modern Food Science and Technology, 2013, Vol.29, No.11, 2566-2571.

To enhance photostability of Monascus pigment, glycinin was modified via glutamyl endopeptidase (GE), which had strict substrate specificity. The surface hydrophobicity of the 11S globulin after hydrolysis was analyzed, and protein/pigment complex was produced after enzymatic modification of the 11S. The binding properties (binding constant and binding capacity) and the light stability of protein/pigment complex were investigated. The results showed that GE had a significant effect on 11S surface character and Monascus pigment photostability. Surface hydrophobicity of 11S increases with the degree of hydrolysis (DH) increased after GE modifying as the inner hydrophobic groups exposed. At pH 7.0, the binding constants of Monascus pigment and 3% (m/V) 11 S reached the maximum of 215.00 U/g pro when the DH was 1.50%. After 24 h light exposure, the protein/pigment complex had the color value retention rate of 90% which highly improved the light stability of Monascus pigment.

Author Response

Response to the fourth reviewer's questions

Interesting paper. The Authors used many methods to make full characterization of obtained complexes.

English language level is satisfactory.

Change the title on:

  1. Preparation, multispectroscopic characterization, and stability analysis of Monascus red pigments and whey protein isolate complex

Answer: Thank you very much for your helpful suggestion. We have changed the title marked by blue color (Line 3-4).

  1. One important paper is missing and it should be cited in the Discussion.

YANG Xiao-quan, YAN Jiang-yin, HOU Jun-jie. Effect of Enzymatic Modification on Binding Capacity between Pigment and Glycinin. Modern Food Science and Technology, 2013, Vol.29, No.11, 2566-2571.

To enhance photostability of Monascus pigment, glycinin was modified via glutamyl endopeptidase (GE), which had strict substrate specificity. The surface hydrophobicity of the 11S globulin after hydrolysis was analyzed, and protein/pigment complex was produced after enzymatic modification of the 11S. The binding properties (binding constant and binding capacity) and the light stability of protein/pigment complex were investigated. The results showed that GE had a significant effect on 11S surface character and Monascus pigment photostability. Surface hydrophobicity of 11S increases with the degree of hydrolysis (DH) increased after GE modifying as the inner hydrophobic groups exposed. At pH 7.0, the binding constants of Monascus pigment and 3% (m/V) 11 S reached the maximum of 215.00 U/g pro when the DH was 1.50%. After 24 h light exposure, the protein/pigment complex had the color value retention rate of 90% which highly improved the light stability of Monascus pigment.

Answer: Thank you very much for your helpful suggestion. We have added this literature to the text with marked by blue color marked by red color (Line 156 and 291).

Reviewer 5 Report

This study investigated the stability of Monascus red pigments with whey protein. Monascus red pigments has already been used as a stable dye industrially. Although the introduction states that the dye is unstable, it exaggerates the instability of the compound for the paper. This is not proper statement. It is not clear that authors imaged what kind of utilization of pigments. There are almost no foods that need to be heated to 100°C for 5 hours even in the sterilization process. In addition, it also seems like there are almost no effect of stability by WPI. What is the unit U? What is m/v? What is Table 1 “a”? Figure 2 I don't understand the difference from B to F. Fig. 3A, authors indicated peaks from pigment. Fig. 5C What is happening at pH 4 and 5? Did authors measure turbidity protein-dye precipitation? What happens to the pigment when the protein precipitates?

Author Response

Response to the fifth reviewer's questions

  1. This study investigated the stability of Monascus red pigments with whey protein. Monascus red pigments has already been used as a stable dye industrially. Although the introduction states that the dye is unstable, it exaggerates the instability of the compound for the paper. This is not proper statement. It is not clear that authors imaged what kind of utilization of pigments. There are almost no foods that need to be heated to 100°C for 5 hours even in the sterilization process. In addition, it also seems like there are almost no effect of stability by WPI.

Answer: Thank you very much for your helpful suggestion. We have revised the introduction marked by green color. Although Monascus red pigments (MRPs) have been used industrially as a stable dye, in some adverse environments its color retention rate can be drastically reduced. We wanted to investigate the color retention rate of MRPs in extreme environments, which is why we chose 100℃. The color retention rate of complex (DH = 0.0%) and complex (DH = 0.5%) was higher than that of MRPs in a water bath at 6090℃.

  1. What is the unit U?

Answer: Thank you very much for your professional question. U is the color value unit of MRPs and color value is one of the main quality indicators of natural pigments [1-3] (Line 120).

  1. What is m/v?

Answer: Thank you very much for your professional question. The m/v is the solute mass to solvent volume ratio. I have used the concentration unit g/L instead of m/v with marked by purple color (Line 120 and 137).

  1. What is Table 1 “a”?

Answer: Sorry for our unclarity expression. Values in the same column with different superscript letters are significantly different (P 0.05) (Line 260, 277  and 435).

  1. Figure 2 I don't understand the difference from B to F.

Answer: Thank you very much for your professional question. Figure 2 shows the SDSPAGE profile of WPI solutions with different hydrolysis degrees. B to F denote WPI with 0.00%, 0.50%, 1.00%, 1.50% and 3.00% degrees of hydrolysis respectively. As you can see from the SDSPAGE profile, WPI with different degrees of hydrolysis have different distributions of protein molecular weight. The results show that different degrees of hydrolysis have an effect on distribution of protein molecular weight.

  1. 3A, authors indicated peaks from pigment.

Answer: Thank you very much for your professional question. As shown in Figure 3A, there is a characteristic peak of pigment at 1640–1670 cm-1, corresponding to the stretching vibration of the carbonyl group (C=O) (Line 319-323).

  1. 5C What is happening at pH 4 and 5? Did authors measure turbidity proteindye precipitation? What happens to the pigment when the protein precipitates?

Answer: Thank you very much for your professional question. The isoelectric point of WPI is pH 4–5, so when the solubility of the protein reaches the isoelectric point (having a net charge of 0) at pH 4 and pH 5, the protein is prone to aggregation and precipitation, so complex (DH = 0.0%) and complex (DH = 0.5%) reach the isoelectric point at pH 4 and pH 5 at 500 nm The absorbance values at 500 nm were too high (>1), resulting in the accumulation and precipitation of both groups of samples, with color value retention of >1.

References

  1. Bai, J.; Gong, Z.; Shu, M.; Zhao, H.; Ye, F.; Tang, C.; Zhang, S.; Zhou, B.; Lu, D.; Zhou, X.; Lin, Q.; Liu, J. Increased Water-Soluble Yellow Monascus Pigment Productivity via Dual Mutagenesis and Submerged Repeated-Batch Fermentation of Monascus purpureus. Front Microbiol. 2022, 13, 914828. https://doi.org/10.3389/fmicb.2022.914828.
  2. Moghadam, H. D.; Tabatabaee Yazdi, F.; Shahidi, F.; Sarabi-Jamab, M.; Es'haghi, Z. Co-culture of Monascus purpureus with Saccharomyces cerevisiae: A strategy for pigments increment and citrinin reduction. Biocatalysis and Agricultural Biotechnology. 2022, 45, 102501. https://doi.org/10.1016/j.bcab.2022.102501.
  3. Hu, Y.; Zheng, Y.; Liu, B.; Gong, Y.; Shao, Y. Mrhst4 gene, coding for NAD+-dependent deacetylase is involved in citrinin production of Monascus ruber. J Appl Microbiol. 2023, 134. https://doi.org/10.1093/jambio/lxad042.

Round 2

Reviewer 2 Report

The paper can be accepted for publication.

Reviewer 3 Report

The manuscript was improved, but requires further changes. The manuscript should be carefully check to apply for the following correction:

Lines 18-19: “MRPs were combined with WPI and 0.5% hydrolysis degree (DH) of WPH to form the complexes “ should be changed to “MRPs were combined with WPI and WPH with hydrolysis degree (DH) of 0.5% to form the complexes …” Otherwise one should understand that the MRP is combined with the hydrolysis degree. This is just an expel. This misleading expression appears in other places as well. The manuscript requires further improvement of the English language.

The Materials and methods section should be carefully checked. In many places the information is delivered as in a protocol. The style should be adapted as for a research paper to be published in a journal.

Reviewer 5 Report

The revised MS was improved. 
